# Heterogeneity of Ex Vivo and In Vivo Properties along the Length of the Abdominal Aortic Aneurysm

Arianna Forneris [1,2] , Miriam Nightingale [1,3], Alina Ismaguilova [1,3], Taisiya Sigaeva [4], Louise Neave [1,3], Amy Bromley [5], Randy D. Moore [6] and Elena S. Di Martino [1,2,3,*]

[1] Biomedical Engineering Program, University of Calgary, Calgary, AB T2N 1N4, Canada; arianna.forneris@ucalgary.ca (A.F.); miriam.nightingale@ucalgary.ca (M.N.); aismagui@ucalgary.ca (A.I.); louise.neave@ucalgary.ca (L.N.)
[2] Department of Civil Engineering, University of Calgary, Calgary, AB T2N 1N4, Canada
[3] Libin Cardiovascular Institute of Alberta, University of Calgary, Calgary, AB T2N 1N4, Canada
[4] Department of Systems Design Engineering, University of Waterloo, Waterloo, ON N2L 3G1, Canada; tsigaeva@uwaterloo.ca
[5] Department of Pathology and Laboratory Medicine, University of Calgary, Calgary, AB T2N 1N4, Canada; Amy.Bromley@albertaprecisionlabs.ca
[6] Department of Surgery, University of Calgary, Calgary, AB T2N 1N4, Canada; rmoor@ucalgary.ca
\* Correspondence: edimarti@ucalgary.ca

**Abstract:** The current clinical guidelines for the management of aortic abdominal aneurysms (AAAs) overlook the structural and mechanical heterogeneity of the aortic tissue and its role in the regional weakening that drives disease progression. This study is a comprehensive investigation of the structural and biomechanical heterogeneity of AAA tissue along the length and circumference of the aorta, by means of regional ex vivo and in vivo properties. Biaxial testing and histological analysis were performed on ex vivo human aortic specimens systematically collected during open repair surgery. Wall-shear stress and three-dimensional principal strain analysis were performed to allow for in vivo regional characterization of individual aortas. A marked effect of position along the aortic length was observed in both ex vivo and in vivo properties, with the central regions corresponding to the aneurysmal sac being significantly different from the adjacent regions. The heterogeneity along the circumference of the aorta was reflected in the ex vivo biaxial response at low strains and histological properties. Present findings uniquely show the importance of regional characterization for aortic assessment and the need to correlate heterogeneity at the tissue level with non-invasive measurements aimed at improving clinical outcomes.

**Keywords:** abdominal aortic aneurysm; biaxial testing; mechanical properties; in vivo strain; wall shear stress; inflammation; regional variations; heterogeneity



## 1. Introduction

An abdominal aortic aneurysm (AAA) is a slowly progressing disease that affects the wall of the abdominal aorta and results in the asymptomatic enlargement of the artery until rupture, which is associated with high mortality. The clinical management of aortic aneurysms is challenging and mostly based on an assessment aimed at weighing the risk of surgery-related complication versus the risk of catastrophic rupture. The maximum aortic diameter (greater than 5 cm) is considered the indicator for elective aortic repair; however, this approach has been proved to lead to suboptimal patient prioritization resulting in many cases of critical aneurysms left untreated or, in contrast, unnecessary interventions on stable aneurysms.

In recent years, different studies highlighted the inadequacy of the diameter criterion by pointing out the high level of heterogeneity in the aneurysmal tissue and its impact on the risk of rupture of individual aortas [1–5]. Aneurysm initiation and progression are multifactorial processes linked to the tissue's heterogeneous remodeling and structural

degradation in response to the local environment, including both mechanical (altered hemodynamics) and biological factors (inflammation, presence of intraluminal thrombus) [6–8]. Considering only the diameter as a metric for rupture risk fails to capture the localized structural weakening and decrease in wall strength that drive aneurysm growth and rupture in individual aortas. In this context, the aortic diameter provides clinicians with limited information and ultimately does not account for inter- and intra-patient heterogeneity. It is therefore essential to fully characterize the local structural and mechanical changes in the aneurysmal tissue with respect to disease progression and rupture potential. This will assist in improving the aortic assessment for clinical purposes through correlation of the heterogeneity at the tissue level with non-invasive measurements.

Uniaxial tensile tests have been used to study the mechanical behaviour of aortic tissue ex vivo [9–11]. However, planar biaxial tests better represent the in vivo loading conditions of the artery and allow for a more appropriate characterization of the three-dimensional mechanical response of the aorta given tissue anisotropy and coupling of fibers in two orthogonal directions. Due to the specifics of AAA-related interventions resulting in minimal tissue being excised, as well as the overall AAA tissue fragility, there is a limited number of biaxial studies in the area [12–15]. Even less studies are concerned with heterogeneity of the AAA wall, with the focus being towards the heterogeneity along the circumference of the aorta [16]. Axial heterogeneity (along the length of the aorta) of ex vivo mechanical properties is absent in the literature. Therefore, a comprehensive assessment of the heterogeneity in both circumferential and axial directions is necessary to fully understand the localized changes in the aneurysmal wall.

As the mechanical behavior of the aorta depends heavily on its microstructure, the assessment of microstructural components in the aortic wall—in terms of content as well as architecture and organization—is essential to fully understand the pathological remodeling associated to disease progression and material properties change. While a reported increase in the collagen-to-elastin ratio is thought to alter the wall structure and thus alter its mechanical response, there is no literature data investigating the heterogeneity of the aortic wall composition along the length and circumference of the aorta [2,17,18].

Immunohistochemical (IHC) analysis provides important information on cellular mechanisms of AAA development and progression. Inflammation and macrophage infiltration are thought to have a significant role in the progressive degradation and remodeling of the extracellular matrix [19]. The primary cells involved in AAA inflammatory response are T-cells and macrophages [20]. Researchers have correlated the distribution of these cells with aneurysm rupture site, intra-luminal thrombus development, and overall disease severity [21]. IHC analysis has been used previously to investigate the regional distribution of inflammatory cells in AAA tissue [16]. However, only circumferential heterogeneity has been reported.

The invasive assessment of the aortic tissue, although central in improving the understanding of disease progression and aortic rupture, represents the first step towards the improvement of clinical guidelines for an accurate, patient-specific evaluation of rupture risk. Equally important is the need to obtain non-invasive means to access information on the state of individual aortas. To this effect, there have been several efforts to model the biomechanics of AAAs, both fluid and solid, that have led to so-called biomechanics-based indices as a surrogate measure to estimate growth and rupture risk [4,5,22,23]. The heterogeneity along the length and circumference of the aorta should be incorporated in these parameters.

In the current study, biaxial testing and histological analysis were performed on human aortic specimens ex vivo. Tissue samples were obtained from a population of open surgical repair patients in order to characterize the heterogeneity of mechanical properties and inflammatory processes of the aneurysmal tissue along the length and circumference of the aorta. Similarly, a series of non-invasive in vivo image-based parameters were obtained along the length and circumference of each aneurysm, namely computational fluid dynamics-based wall-shear stress and three-dimensional principal strain. Hence,

this work provides a comprehensive investigation of the structural and biomechanical heterogeneity of the abdominal aortic aneurysm.

## 2. Materials and Methods

### 2.1. Patients and Aortic Tissue Samples

Aortic tissue samples were obtained from a population of AAA patients who consented to participate in the study between 2016 and 2020. The research protocol, approved by the University of Calgary Conjoint Health Research Ethics Board (Ethics ID: REB15-0777) included pre-operative electrocardiography-gated dynamic computed tomography (CT) imaging and open surgical aneurysm repair with complete aortic resection that allowed for the collection of specimens from different regions (i.e., aneurysm sac, aneurysm neck, anterior, posterior, left, right) of individual aortas. A sampling grid with 24 regions (or patches) was utilized in the systematic harvesting of aortic samples. Patches were defined perpendicularly to the lumen centerline of each aortic geometry reconstructed from the first phase of the dynamic CT scans (Figure 1a). Specifically, the 24 regions resulted from the subdivision of each aortic geometry into 6 regions along the length of the vessel ('1'—neck, '6'—bottom; average patch length along the centerline was $20 \pm 4$ mm) and 4 regions along the circumference ('L' and 'R'—left and right sides of the aneurysm, 'P' and 'A'—anterior and posterior sides of the aneurysm with respect to the geometry centerline). Both the lumen and outer aortic wall were reconstructed by means of image processing and semi-automatic, threshold-based image segmentation with the imaging software Simpleware ScanIP (Synopsys Inc., Mountain View, CA, USA). The use of a sampling grid was central to the study as it facilitated tracking of specimen locations and regional analysis of both ex vivo and in vivo properties.

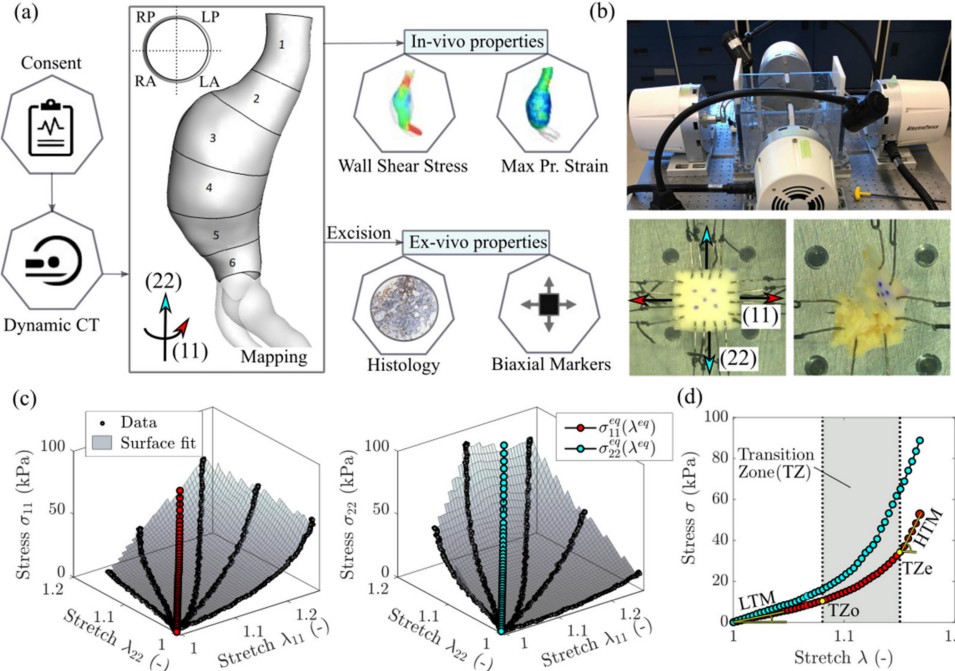

**Figure 1.** (**a**) Methodology summary and demonstration of the sampling grid defining the circumferential regions (RP—right posterior, LP—left posterior, RA—right anterior, LA—left anterior) and axial regions (1 to 6) of an aneurysm. (**b**) Biaxial testing setup and representative specimens: normal-sized specimen with four hooks per side and smaller specimen with two hooks per side. The latter is shown to fail due to tissue fragility. (**c**) A 3D representation of the biaxial data for a displacement-controlled test with different loading ratios. The surface interpolation is used to determine the equi-biaxial stretch-stress state. Stress and stretch are defined in the circumferential direction (11) and the longitudinal direction (22). (**d**) A 2D representation of the equi-biaxial stretch state and determination of all relevant mechanical properties—the low and high tangential modulus (LTM and HTM), the transition zone onset and end (TZo and TZe).

## 2.2. Ex Vivo Analysis

### 2.2.1. Biaxial Testing and Data Analysis

Biaxial testing was carried out on a four-motor biaxial testing system (ElectroForce Systems, TA Instruments, Springfield, MO, USA, Figure 1b), designed to allow independent control of each motor. Full-thickness tissue samples were cut into squares (target dimensions—10 × 10 mm) and mounted to the four linear motors via sutures and hooks (Figure 1b). Four hooks per side were used to minimize edge effects and ensure homogeneous strain distribution at the center of specimens [24]. In the case of samples with dimensions too small to accommodate four hooks, two hooks per side were used as recommended in Slazansky et al., 2016 [24]. Samples were carefully oriented with direction (11) corresponding to the direction along the circumference of the aorta (circumferential direction), and direction (22) along the length of the aorta (longitudinal direction). The thickness and side-to-side distances (where sides were assumed along the points of hooks attachment) were measured using a caliper. Five dots were drawn on the sample central region using a surgical skin marker to provide distinguishable marks that can be tracked by the overhead camera. A high-resolution digital video extensometer (DVE) camera (640 × 480 pixel resolution, 55 m focal length, 200 frames per second) mounted above the test specimen was used to track the dots, producing deformation measures at the central region of the sample. When testing a tissue's response, the in vivo environment must be mimicked. As such, the sample was fully immersed in a (PBS) solution at 37 °C and pH 7.4.

Prior to testing, a pre-load of 0.05 N was applied to avoid sagging effects due to sutures slack. Loading of the sample was recorded using two load cells (22 N). At first, due to AAA tissue fragility and a high risk of premature tearing, displacements resulting in a simultaneous 20% increase of the distances between parallel sides of the sample were assigned. Then, to achieve a wider range of mechanical responses, the sample was extended with different ratios of these initial displacements. Next, the procedure was repeated for larger loadings, namely 40% and 60%. For each test, samples were conditioned by running the test for 5 cycles at a rate of 0.6 mm/s. Over 30% of the specimens failed prematurely due to disruption of tissue by hook tear-through or delamination (Figure 1b). For the surviving specimens, the data was collected from the last cycles of the tests associated with the 40% displacement protocol.

The principal stretches $\lambda_{11}$ and $\lambda_{22}$ were calculated from the deformations of the center of the specimen enclosed by the tracked markers [25]. The Cauchy stresses $\sigma_{11}$ and $\sigma_{22}$ were determined from the forces required to displace the specimen's sides along two directions multiplied by the corresponding principal stretch and divided by the undeformed area (i.e., the corresponding side length times thickness). The three-dimensional stress–stretch curves derived from the biaxial tests for a representative specimen are shown in Figure 1c. We specify that the assigned displacements do not necessarily produce equivalent deformation data points ($\lambda_{11}$, $\lambda_{22}$) for different specimens because we do not control $\lambda_{11}$:$\lambda_{22}$ ratios [25]. In order to achieve mutual comparison between specimens, one option is to apply constitutive modelling and produce comparable stress–stretch data points, for which different effective mechanical properties such as stiffness and transition strains could be determined. To bypass the constitutive modelling stage, we reached the same result by fitting a surface to the data in the three-dimensional space and interpolating the stresses $\sigma_{11}^{eq}$ and $\sigma_{22}^{eq}$ at the equi-stretch deformation state (i.e., $\lambda_{11} = \lambda_{22} = \lambda^{eq}$). This analysis was conducted in MATLAB (version R2018a; MathWorks, Natick, MA, USA). The equi-biaxial mechanical response for a representative dataset is shown in Figure 1 in both 3D (Figure 1c) and 2D (Figure 1d) versions. Repeating the procedure for all the specimens allows comparison of the tissues' mechanical properties at an equivalent deformation state. This approach represents an attractive alternative to constitutive modelling when finite element simulations are not the end goal.

The curves obtained from testing were then used to calculate the mechanical properties using a custom software pipeline developed in Python (version 3.7.6; Python Software Foundation). All mechanical parameters are shown in Figure 1d for an example of unpro-

cessed equi-biaxial stress-stretch response in the circumferential direction (11). Data was first processed using a Savitsky–Golay digital filter, which relies on a local least-squares polynomial approximation to reduce noise without distorting the signal [26]. The linear regions of the curves can be described with a low-strain tangential modulus (LTM) and high-strain tangential modulus (HTM) which characterize the elastin-based and collagen-dominated response of the tissue, respectively [27,28]. These two linear regions are separated by a non-linear transition zone where the load is shared by varying proportions of collagen and elastin (Figure 1d). To determine LTM, HTM, and the transition zone, a polygonal approximation of the curve was generated using a Ramer–Douglas–Peucker (RDP) algorithm [29] with a fixed deviation value of 0.025 to produce a high-precision fit. LTM and HTM were evaluated as the slopes of the first and last segment, respectively, while the transition zone onset (TZo) and transition zone end (TZe) were found as the start and end points of the transition zone between the LTM and HTM segments (Figure 1d). For curves with two segments or less, there was deemed to be no transition zone. A visual representation of RDP and original curves were overlaid to confirm fit. Given the marked difference in AAA mechanical behavior with respect to direction, the level of mechanical anisotropy was also investigated for each mechanical parameter (MP) extracted from the biaxial response of the tissue samples as the ratio of the response in the circumferential direction to the response in the axial direction ($MP_{11}/MP_{22}$).

### 2.2.2. Histological Analysis

Histological analysis involving Musto/Movat pentachrome staining to evaluate elastin, proteoglycans, and smooth muscle cells content was performed at the Calgary Laboratory Services (Core Pathology Laboratory). Aortic specimens were fixed with 10% buffered formaldehyde solution immediately after collection, embedded in paraffin on edge and cut at 4 μm sections. Each stained sample was then analyzed by means of colorimetric analysis through the Aperio ImageScope software (version 12.3; Leica Biosystems Inc., Concord, ON, Canada). A positive pixel algorithm, validated through the Core Pathology Laboratory, was used to detect the color black, cyan, and magenta to assess the relative area of elastin, proteoglycans and vascular smooth muscle cells, respectively, and estimate their content in the specimens. The results of the colorimetric analysis was reported as a stained area ($mm^2$) within the total analyzed region ($mm^2$).

### 2.2.3. Immunohistochemical Analysis

IHC analysis involving immunostaining was performed at the Calgary Laboratory Services (Core Pathology Laboratory). Aortic tissue samples were fixed with 10% buffered formaldehyde solution upon harvesting in the operating room, embedded in paraffin, on edge, and cut at 4 μm sections. A Dako EnVision Flex Detection Kit (Agilent Technologies, Santa Clara, CA, USA) was used for the evaluation of inflammatory cells; specifically, the presence of the following markers was quantified: helper T-cells (CD4+), cytotoxic T-cells (CD8+), and macrophages (CD68+) [4]. This ready to use kit was used to the manufacturer's specifications (no dilution or secondary antibody retrieval necessary) with batch controls (tonsil). Analysis for CD4 (SP35) was run with H15 × 15 (15 min antibody incubation, 15 min detection kit incubation) at 1/25, for CD8 (C8/144B) with H20 × 20, and for CD68 (KP1) with H15 × 20. IHC analysis was performed in order to characterize and quantify the level of inflammation in the media and adventitia layers as the mean number of cells per 1 $mm^2$ specimen area. The immunostaining was counterstained with hematoxylin allowing the discrimination of the adventitia layer from the media layer. The presence of each marker was evaluated by manually counting the positively stained cells in a 1 $mm^2$ area identified as a hot-spot with the highest density of stained cells.

### 2.3. In Vivo Analysis

A series of in vivo analyses were performed to characterize each patch defined on the aortic geometries and allow for the correlation of in vivo and ex vivo measures on

corresponding regions. For this reason, each descriptor derived from the in vivo analysis was obtained as a continuous distribution on the aorta's surface geometry as well as region-averaged distribution on the 24 regions defined on each geometry as an intra-operative sampling grid.

### 2.3.1. Wall Shear Stress

The evaluation of the wall shear stress was performed by means of computational fluid dynamic (CFD) simulations in order to characterize the local fluid dynamic patterns and quantify local hemodynamic disturbances. The reconstructed geometry of the aortic lumen for individual aortas was imported in Icem (version 2019.2; Ansys, Canonsburg, PA, USA) and discretized into tetrahedral elements with a final mesh density of approximately 2 to 3 million elements selected following a mesh sensitivity analysis. The use of a boundary layer consisting of prismatic elements allowed for an improved result accuracy at the geometry boundary where the variable of interest, the wall shear stress, was computed. The CFD commercial software Fluent (version 2019.2; Ansys, Canonsburg, PA, USA) was used to run transient-time simulations and reproduce the aortic fluid dynamics during a cardiac cycle. A semi-implicit method for pressure-linked equations (SIMPLE algorithm) was employed to couple pressure and velocity and solve the Navier–Stokes equations for the blood modelled as an incompressible, Newtonian fluid undergoing laminar flow [4,5]. A second order upwind formulation and a second order implicit transient formulation were used for spatial and temporal discretization respectively. Boundary conditions were imposed in order to enable a solution of the fluid flow equations. Specifically, a velocity boundary condition was defined at the computational domain inlet [30], an outflow boundary condition was prescribed at the outlets with 50% flow division into each of the iliac arteries and a no-slip condition was imposed at the wall of the fluid domain (fluid interface) assumed as rigid. The results of CFD simulations were post-processed to derive the time-averaged wall shear stress (TAWSS) representing the shear-stress loading on the aortic wall averaged over the cardiac cycle as expressed by the equation

$$\text{TAWSS} = \frac{1}{T} \int_0^T |\text{WSS}(s,t)| \, dt, \tag{1}$$

where T is the duration of the cardiac cycle and WSS(s,t) is the magnitude of the wall shear stress vector at a specific location (s) and time (t).

### 2.3.2. Maximum Principal Strain

Three-dimensional principal strain analysis was performed in vivo on dynamic CT images by means of proprietary software ViTAA (Virtual Touch Aortic Aneurysm—patent WO-2018/068153-A1) [31,32]. The strain analysis relied on an optical flow-based algorithm and used a triangular surface mesh of the outer aortic wall as a 3D tracking model in order to measure the displacement of each mesh node over the cardiac cycle and derive the deformation gradient. Finally, the in vivo principal strain was computed from the Green-Lagrange tensor [31,32].

### 2.4. Statistical Analysis

The statistical analysis was completed using a custom software pipeline developed using the SciPy, Pingouin, and Statsmodels packages in Python [33–35]. The normality of all continuous variables was assessed using the Shapiro–Wilk test. Variables that did not pass an initial significance threshold of 0.05, were subjected to a Box–Cox transformation. Normality was also assessed visually through histograms and QQ (quantile-quantile) plots. The equivalence of variance for each continuous variable along position (categorical) was assessed through the Levene test with a threshold of 0.05. A linear mixed effect model was used to assess the relationships presented in this study, with inter-patient variability set as a random effect. The significance threshold was set to 0.05 for all models. Due to the robustness of these models to violations of the distributional assumptions, the

significance for the Shapiro–Wilk test after the Box–Cox transformation was set to 0.1 [36]. The normality and heteroscedasticity of the residuals of all linear mixed effect models were assessed through the Shapiro–Wilk and the White test respectively.

## 3. Results

A total of 12 patients referred to elective AAA open repair surgery gave consent to be enrolled in the study (mean age $65 \pm 6$ years, 92% males). Hypertension was reported in 7 patients in the population (58%). Four patients (33%) presented dyslipidemia, with one patient also presenting coronary artery disease, while two patients (17%) in the population presented peripheral vascular disease.

The average maximum diameter for the population was $4.94 \pm 0.85$ cm, with 4 patients (33%) selected for surgery because they presented critical complications (e.g., penetrating atherosclerotic ulcer, acute ischemic leg, rapid aortic growth, peripheral vascular disease) despite having a maximum aortic diameter lower than 5 cm (i.e., current clinical threshold used for surgical decision). The study protocol included open repair procedure with complete aortic resection resulting in the aorta not being opened during surgery with consequent minimal blood loss for the patients. The procedure led to an average length of stay of 7.7 days for the patients, consistently with the average length of stay reported for cohorts receiving open repair surgery without complete aortic resection.

The shape of the AAA varied across patients. Position 1 was associated with the neck for all patients and exhibited no dilatation; this region was considered a non-dilated tissue control in the longitudinal heterogeneity analysis. Position 2 exhibited no dilatation in 5/12 patients with the remaining patients having some dilatation beginning in this area. Position 3, 4, and 5 were the central regions of the aneurysm and exhibited the most growth with the largest dilatation being present in position 3 for 6/12 patients, position 4 for 4/12 patients, and position 5 for 4/12 patients. In some cases where the maximum dilatation spanned multiple regions, both positions were accounted for. For position 6, 6/12 patients exhibited a decrease in dilatation while for the rest of the patients, this area remained largely dilated. A double aneurysm, where two separate dilatations are present along the abdominal aorta, was seen in 3/12 patients, with the first generally present around position 3 and another area of growth at position 5 and 6.

The results reported below concern the changes of selected ex vivo and in vivo properties along the length and circumference of the aorta. Only statistically significant results are reported and discussed.

### 3.1. Ex Vivo Analysis

3.1.1. Biaxial Mechanical Response of the Aortic Tissue

With approximately 30% of the specimens failing due to rupture during experimental set-up or during the initial testing protocol before reaching the 40% displacement protocol, a complete biaxial test was feasible for 24 aortic samples from 7 aneurysms. The average LTM characterizing the linear region of the tissue's response for the analysed specimens was $298.3 \pm 267.3$ KPa and $359.8 \pm 380.7$ KPa in the circumferential (11) and longitudinal (22) direction respectively, with an average anisotropic index of $1.73 \pm 4.31$. The average HTM was found to be $3121.0 \pm 1666.1$ KPa in the circumferential direction and $3381.3 \pm 2605.4$ KPa in the longitudinal direction, with an average anisotropic index of $2.27 \pm 5.43$. The tangential modulus at both low and high strain was not significantly different in the circumferential versus longitudinal direction.

The LTM did not present a significant effect of position along the aortic length, but showed a statistically significant effect of position along the circumference (Figure 2a,b). Samples collected from the left regions of the aorta exhibited higher LTM in the circumferential direction ($p = 0.039$ for left-posterior versus right-anterior; $p = 0.061$ for left-posterior versus right-posterior) and higher LTM in the longitudinal direction ($p = 0.044$ for left-anterior versus right-anterior; $p = 0.027$ for left-anterior versus right-posterior).

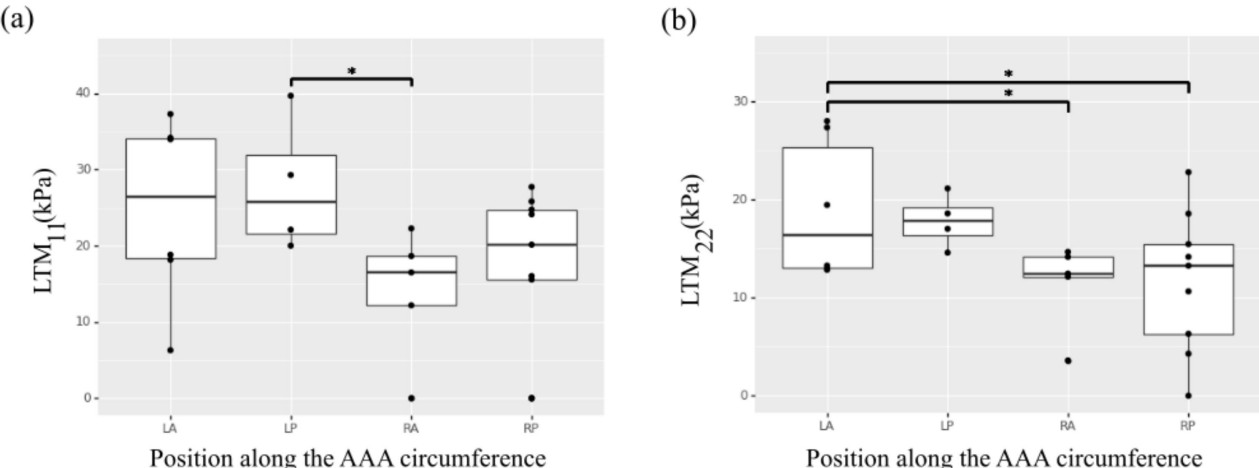

**Figure 2.** Heterogeneity of the selected ex vivo biaxial mechanical properties along the circumference of the AAA. (**a**) Low tangential modulus (LTM) in the circumferential direction (11). (**b**) Low tangential modulus (LTM) in the longitudinal direction (22). * indicates *p* value < 0.05.

Of note, a statistically significant effect of position along the aneurysm length was observed for the circumferential HTM (Figure 3a). In particular, the aneurysm neck region in position 1 exhibited higher circumferential HTM compared to the aneurysmal sac regions corresponding to position 3 (*p* = 0.001), position 4 (*p* = 0.053), and position 5 (*p* = 0.010). An effect of position was also observed for the HTM anisotropic index. The index was transformed through an exponential function to obtain a normal distribution. Thus, a positive number indicates a preferred fiber directionality in the circumference direction while a negative number is associated with axially oriented fibers. As these numbers approach 0, the material behaves more isotropically. The samples collected from regions in position 3 showed lower anisotropy and reached statistical significance when compared to the mechanical response characterizing position 2 or 4 (*p* < 0.05) (Figure 3b).

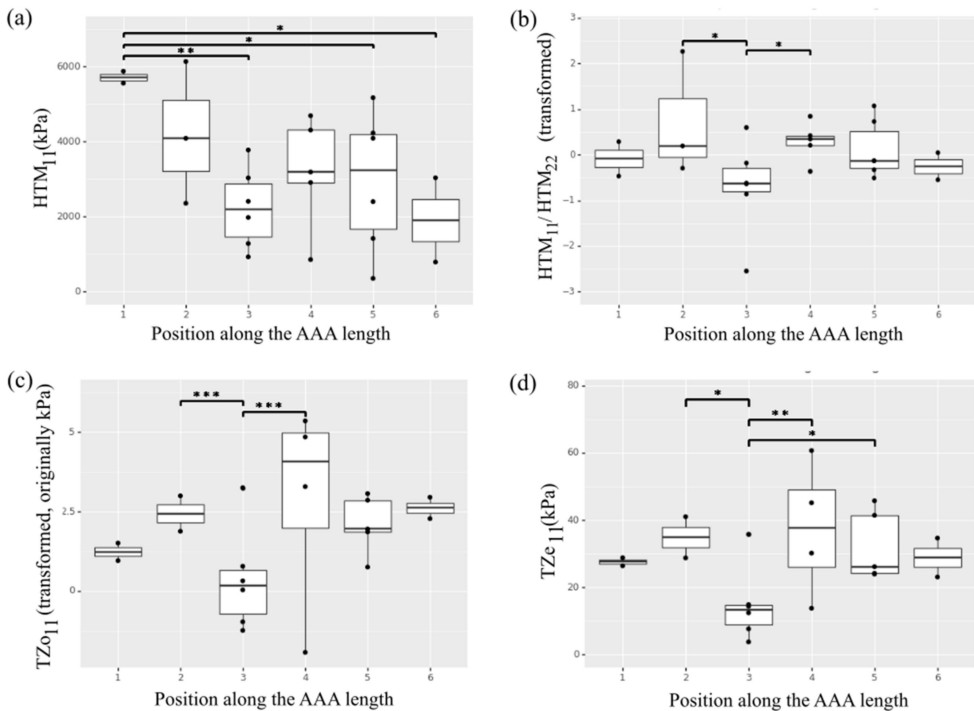

**Figure 3.** Heterogeneity of the selected ex vivo biaxial mechanical properties along the length of the AAA. (**a**) High tangential modulus (HTM) in the circumferential direction (11). (**b**) High tangential modulus anisotropy. (**c**) Transition zone onset (TZo) in the circumferential direction (11). (**d**) Transition zone end (TZe) in the circumferential direction (11). * indicates *p* value < 0.05, ** indicates *p* value < 0.01 and *** indicates *p* value < 0.001.

The non-linear transition zone of the stress-stretch curve for each AAA sample was described by means of TZo and TZe. The average TZo for the analyzed specimens was $5.3 \pm 5.1$ KPa in the circumferential direction and $6.1 \pm 4.6$ KPa in the longitudinal direction, with an average anisotropic index of $2.31 \pm 6.36$. The average TZe was found to be $27.8 \pm 13.8$ KPa and $35.0 \pm 28.4$ KPa respectively in the circumferential and longitudinal direction (average anisotropic index was $1.52 \pm 1.84$). The effect of position along the length of the artery on the parameters describing the transition zone was observed for the circumferential direction (Figure 3c,d), with samples in the central region in position 3 showing lower TZo ($p < 0.001$ compared to position 2 and 4; $p = 0.06$ compared to position 6) and TZe ($p < 0.05$ compared to position 2 and 4; $p = 0.083$ compared to position 6). The parameters describing the transition zone showed no significant effect of position along the aortic circumference.

### 3.1.2. Aortic Media Composition

Histological samples were collected from a subgroup of 9 patients and aortic composition was evaluated by means of Musto/Movat pentachrome staining to identify elastin, proteoglycans, and smooth muscle cells content for a total of 109 samples. Fifty-six samples (51%) had no discernible media layer and had to be excluded. Figure 4 shows a representative histology for two AAA specimens. The average elastin content for the histological specimens was $23.6\% \pm 10.3\%$, average smooth muscle cells content was $55.0\% \pm 16.5\%$ and average proteoglycans content was $21.5\% \pm 6.5\%$.

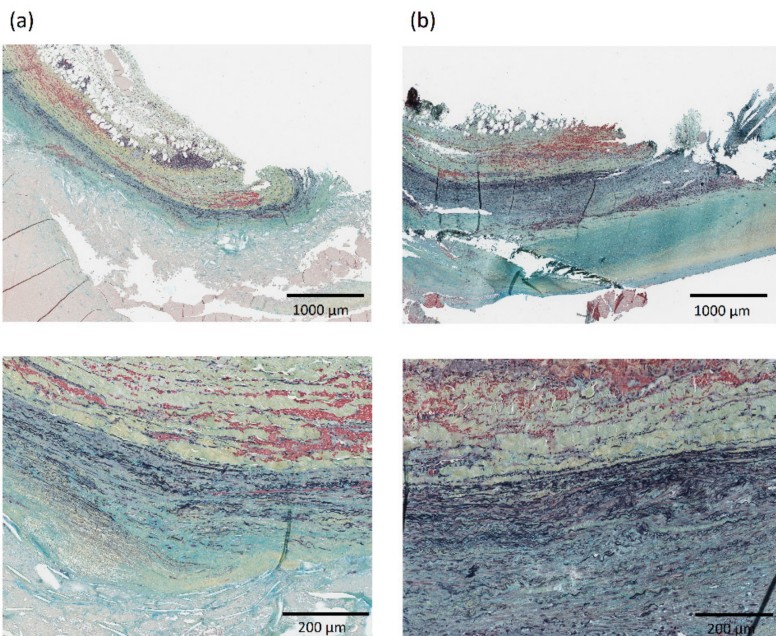

**Figure 4.** Representative histology showing aortic composition visualized with Musto/Movat pentachrome stain in the neck—region RA1 (column **a**) and aneurysmal dilatation—region LA3 (column **b**) for one AAA patient in the population. The layers in each histology from top to bottom are: adventitia, media, and intima. Images at 2× (top) and 10× (bottom) magnification are shown.

The composition of the media layer showed no statistically significant effect of specimen position along the length of the aorta (positions 1 to 6). A significant effect of circumferential position (right-posterior, left-posterior, right-anterior, left-anterior) on elastin, smooth muscle cells and proteoglycans content was observed (Figure 5), with specimens in anterior positions showing lower elastin content, higher smooth muscle cells content and lower proteoglycans content compared to posterior regions. Specifically left-anterior versus left-posterior, left-anterior versus right-posterior, right-anterior versus right-posterior, and

right-anterior versus left-posterior ($p < 0.01$ for elastin and smooth muscle cells contents and $p < 0.04$ for proteoglycans content) (Figure 5).

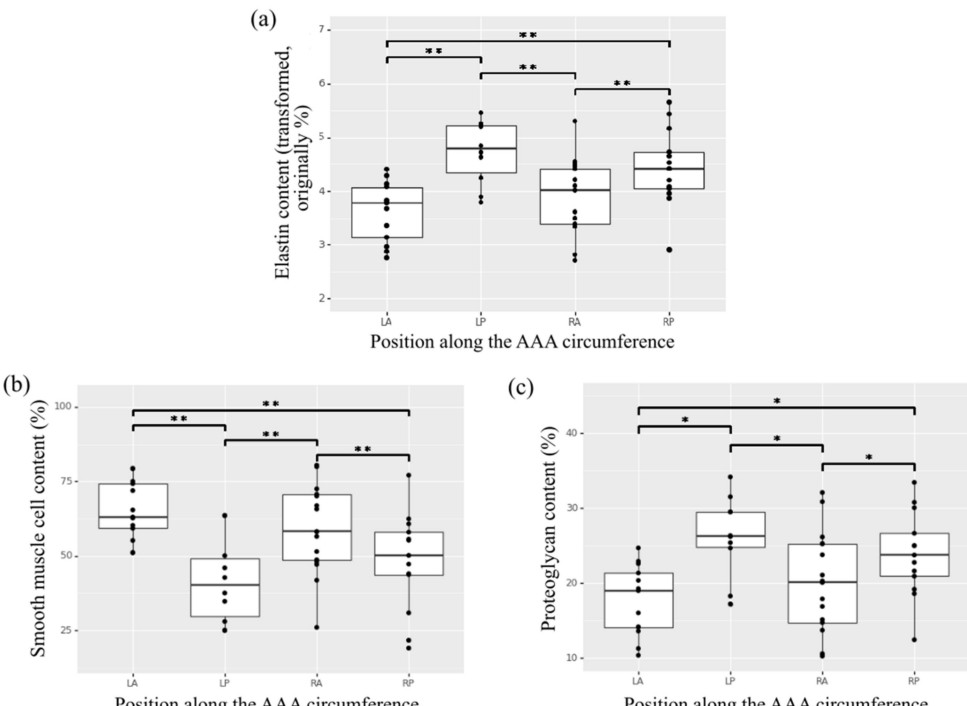

**Figure 5.** Heterogeneity of content along the circumference of the AAA for the aortic wall constituents (**a**) elastin, (**b**) smooth muscle cells, and (**c**) proteoglycans in the media of histological aortic samples. * indicates $p$ value < 0.05, ** indicates $p$ value < 0.01.

### 3.1.3. Markers of Inflammation

A subgroup of 7 patients allowed for the collection of aortic tissue specimens for which immunohistochemical analysis was feasible. Cell count for inflammation was therefore performed on 46 aortic samples, while several specimens had a very thin or absent media layer determining their exclusion from the analysis. The average cell count was 37. $8 \pm 49.4$ cells/mm$^2$ and $183.1 \pm 429.5$ cells/mm$^2$ for the helper T-cells (CD4+) in the media and adventitia layer respectively, $42.0 \pm 49.9$ cells/mm$^2$ and $92.8 \pm 127.3$ cells/mm$^2$ for cytotoxic T-cells (CD8+), and $22.2 \pm 29.1$ cells/mm$^2$ and $45.3 \pm 37.9$ cells/mm$^2$ for the macrophages (CD68+). Figure 6 shows a representative immunostaining image from IHC analysis with inflammatory infiltrate of CD4+, CD8+, and CD68+ cells for two AAA specimens.

Overall, the adventitia layer showed a significantly higher presence of inflammation markers compared to the media layer ($p < 0.05$ for CD4+, $p < 0.05$ for CD8+, $p < 0.01$ for CD68+). The adventitia layer also showed an effect of position observed for the CD4+ and CD8+ markers, with specimens collected from the central regions of the aorta, located in the aneurysmal dilatation, showing higher content of the two inflammatory infiltrates (Figure 7a,b). Of note, the samples collected from region 2 corresponding to the aneurysm neck showed statistically significant lower presence of CD4+ and CD8+ markers in the adventitia layer compared to samples in position 4 ($p = 0.036$ and $p = 0.004$ for CD4+ and CD8+ respectively). No significant differences were observed in the media layer in terms of inflammation and inflammatory infiltrate content with respect to position.

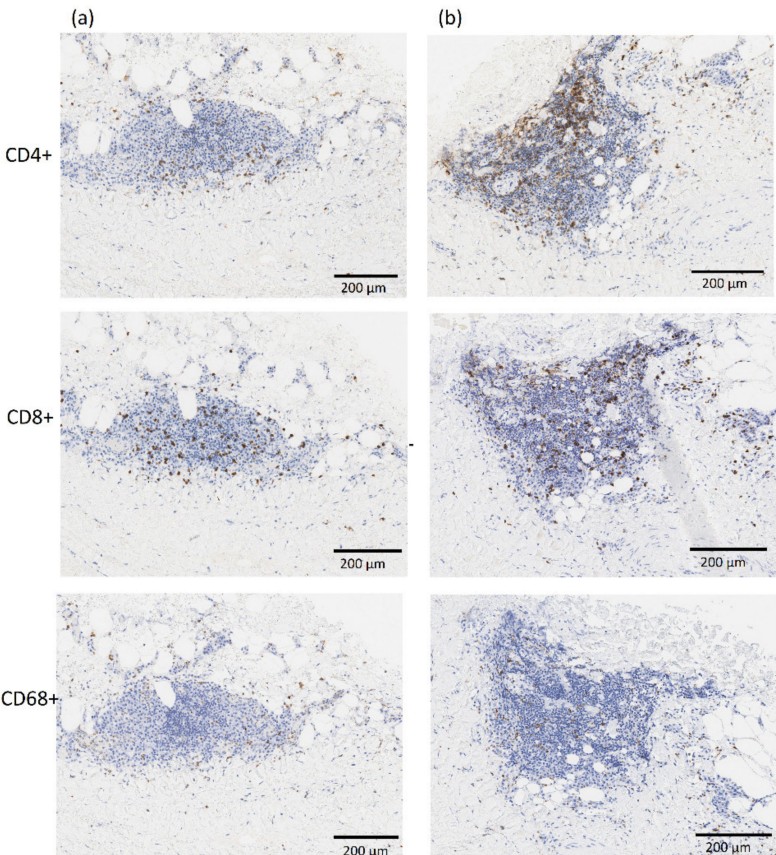

**Figure 6.** Representative images from immunostaining showing inflammatory infiltrate of CD4+, CD8+ and CD68+ cells in the neck—region RA1 (column **a**) and aneurysmal dilatation—region RA3 (column **b**) for one AAA patient in the population. The adventitia and media layers are in the top and bottom half of each image, respectively. All images are at $10\times$ magnification.

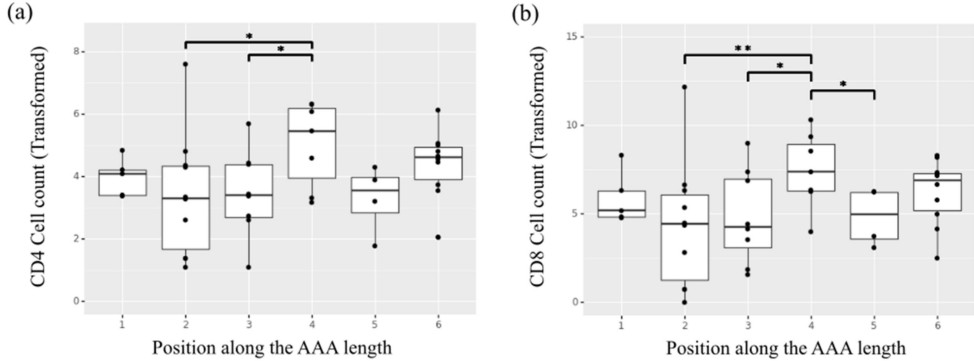

**Figure 7.** Heterogeneity of cell counts along the length of the AAA for the inflammatory infiltrates (**a**) helper T-cells (CD4+) and (**b**) cytotoxic T-cells (CD8+) in the adventitia layer of the selected samples undergoing ex vivo immunohistochemical analysis. * indicates *p* value < 0.05, ** indicates *p* value < 0.01.

### 3.2. In Vivo Analysis

The in vivo analysis was performed to obtain the CFD-based TAWSS distribution on the luminal surface of the aortic geometry and the maximum principal strain distribution derived from dynamic CT images on the outer wall surface of the artery. The in vivo analysis of maximum principal strain and wall shear stresses did not require invasive measurements and was conducted for a larger cohort of aneurysms (n = 12) including the ones selected for the ex vivo analysis. For TAWSS and maximum principal strain, a

region-averaged distribution was obtained in order to characterize the same regions (24 on each aorta) with an average value.

The mean region-averaged TAWSS for the study cohort was $0.50 \pm 0.24$ Pa. A significant effect of position along the length of the aorta for the TAWSS was observed (Figure 8a). The central regions across the aortic geometries—namely, regions 3, 4, 5—presented lower TAWSS resulting from the slow and highly recirculating flow patterns that characterized the aneurysmal dilatation, while regions upstream (aneurysm neck region 1 and 2) and downstream the dilatation (region 6) presented a more organized, high-velocity flow that corresponded to higher TAWSS values. Of note, region 1 was characterized by statistically significant higher TAWSS compared to any other region ($p < 0.001$) while region 2 was characterized by significantly higher TAWSS compared to the more central regions in position 3 ($p = 0.004$) and position 4 and 5 ($p < 0.001$). The same central regions showed significantly lower TAWSS with respect to the areas in position 6 downstream the aneurysm sac ($p = 0.023$ with respect to position 3, $p < 0.001$ with respect to position 4 and 5).

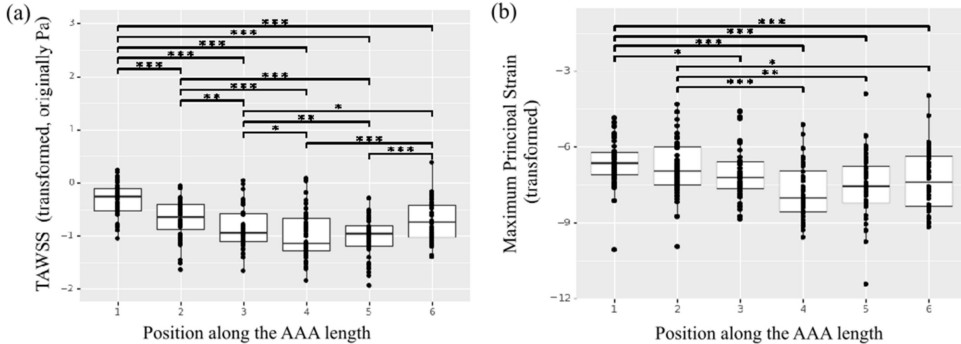

**Figure 8.** Heterogeneity of the in vivo properties along the length of the AAA. (**a**) TAWSS. (**b**) Maximum principal strain. * indicates $p$ value < 0.05, ** indicates $p$ value < 0.01 and *** indicates $p$ value < 0.001.

The in vivo strain analysis performed on dynamic ECG-gated CT images was feasible for 11 patients in the study population, as the scan for one patient did not meet the inclusion criteria due to poor image quality. The mean region-averaged maximum principal strain for the population was $0.03 \pm 0.01$. Once again, the regional analysis pointed to a significant effect of position along the length of the aorta (Figure 8b), with central patches exhibiting smaller strain compared to the aneurysm neck and downstream locations. In particular, the neck region in position 1 presented a significantly larger strain compared to any of the central regions ($p = 0.02$ for region 3, $p < 0.001$ for regions 4 and 5).

Both in vivo variables of interest showed no significant effect of position along the circumference of the aorta.

An interesting relationship was found when comparing the TAWSS for the aortic regions with collected specimens showing no discernible media layer versus discernible media layer. The aortic tissue showing highly compromised composition with no clearly discernible media layer presented statistically significant lower TAWSS ($p < 0.001$) (Figure 9).

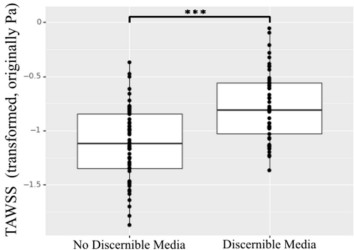

**Figure 9.** TAWSS for specimens showing no discernible media layer versus discernible media layer. *** indicates $p$ value < 0.001.

## 4. Discussion

The present study aimed at investigating the structural and biomechanical heterogeneity of aortic aneurysmal tissue, with respect to the position along the length and circumference of the artery. Both invasive and non-invasive image-based measures were performed on regions of AAAs for a population of patients referred to elective surgery. Specifically, open repair surgery involving complete aortic resection allowed for the collection of tissue specimens from different regions along the aorta, enabling the ex vivo regional characterization of the aneurysmal tissue by means of biaxial testing, with novel biaxial parameters and histological analysis. It is important to note that studies on the ex vivo characterization of aneurysmal tissue often focus on one specimen as representative of the aneurysm or on few samples mostly collected from the anterior portion of the aorta [12]. For this reason, research dealing with regional differences has often been carried out on animal models in order to compensate for the limited availability of human tissue [14,15].

Biaxial mechanical testing of AAA tissue enables accurate assessment of the coupling of fibers in two orthogonal directions—and thus of anisotropy—spanning a range of physiological stresses/strains. However, studies focusing on biaxial testing are scarce due to aneurysmal tissue fragility. In our study, more than 30% specimens failed before reaching the 40% displacement protocol. The remaining specimens required extremely delicate handling.

Alterations in the biomechanical properties of the AAA are attributed to the changes in the structure of the extracellular matrix (ECM) constituents, mainly the elastin and collagen fibers. The low-strain tangential modulus (LTM) is associated with the behavior at low strain often attributed to the elastin component. The onset stress at the beginning of the transition zone (TZo) marks the instant when the collagen begins to engage and resist the applied loading. The end of the transition zone (TZe) is indicative of the collagen in the tissue being fully activated and the transition from the elastin-dominated mechanical response to the collagen-dominated response is complete. Finally, the high-strain tangential modulus (HTM) captures the fully engaged collagen-associated behavior.

The LTM was the only mechanical parameter found to be significantly affected by the position along the circumference of the AAA. The LTM is associated with the elastin-dominated region of aortic tissue behavior. Thus, this asymmetry could be associated with varying distributions of elastin content. In fact, the left patches of AAAs were found to have higher elastin content, which was clearly reflected in the biaxial data showing higher LTM in the left-anterior (LA) and left-posterior (LP) regions. Similarly, histological samples collected from posterior regions showed higher elastin content reflected in the trend towards higher LTM for posterior biaxial specimens. Interestingly, significant differences in LTM were associated with opposing circumferential regions.

Other studies also reported the heterogeneity along the circumferential regions of the aorta [11,37]. However, their assessment was based on uniaxial biomechanical parameters, and thus, their results are true for the high-strain mechanical response only with no consideration for the coupling of fibers in two orthogonal directions. The present study, to the best of the authors' knowledge, is the first to report circumferential heterogeneity at the low strain regime (LTM).

Regarding the axial heterogeneity, there was no effect of the position along the length of the aorta on LTM values, potentially due to the higher variability in LTM, especially in the aneurysmal sac (position 3 and 4).

The present results show that the collagen-related biaxial properties TZo, TZe, and HTM in the circumferential direction were significantly different for tissue samples collected from different regions along the length of the AAA (positions 1 to 6). This key result suggests that the pathology has a variable effect on the collagen fibers—which are oriented predominantly in the circumferential direction—along the different segments of the aneurysm resulting in longitudinal heterogeneity. Literature findings show that aneurysm progression results in the alterations of various biaxial properties along the circumferential

direction (11) of AAAs [12]. However, variation of these circumferential changes along the length of AAA, to the best of the authors' knowledge, have not been addressed.

Despite the lack of studies, the longitudinal heterogeneity of the circumferential response of the tissue may be due to the large fluctuations in diameter along the aortic geometry in the presence of an aneurysmal dilatation. Collagen has a relatively short half-life and newly deposited collagen at locations corresponding to large circumferential dilatation may present lower fiber ondulation [38]. Future studies should explore how collagen fibers change along the length of the AAA.

When compared to other regions along the length of the aorta, the aneurysm neck (position 1) was found to be the stiffest, with significantly higher circumferential HTM (direction 11). The aneurysm neck's tissue could be considered as representative of the non-dilated tissue given its position upstream the enlargement. In this case, we would obtain opposite result with respect to other studies that report the aneurysmal tissue as stiffer compared to a healthy control [12]. In contrast, our results suggest that the presence of an aneurysm leads to a decrease in HTM/stiffness along the length of the aorta as it dilates within the same patient. In fact, position 3 (located in the aneurysmal sac) exhibited both the lowest HTM and the largest diameter within our investigated subset of patients. Previous studies have shown that the collagen present in the aneurysmal tissue is more disorganized, as well as thinner and more elongated, potentially resulting in lower circumferential HTM at sites of advanced disease progression versus non-dilated tissue such as the neck [39].

In contrast to previous studies [12,40], we did not observe a pronounced anisotropy when analyzing the specimens altogether. In particular, the neck (position 1) and the region downstream of the aneurysm (position 6) were observed to be isotropic. However, a significant difference was observed in the anisotropy associated with HTM with respect to the longitudinal position within the body. Position 3 had a tendency towards a preferred longitudinal fiber directionality, while the adjacent patches (position 2 and 4) exhibited preferred fiber directionality in the circumferential direction. The drastic change in anisotropy associated with position 3 indicates the disorganization of circumferentially aligned collagen fibers in the aneurysmal sac resulting in a lower circumferential HTM as seen in this study, but little change in longitudinal HTM. The heterogeneity of the tissue's mechanical response may explain the inconsistency between the present results and other studies, in which only one sample was excised from the aneurysm and, thus, was representative of only one region.

The infamous position 3 (in the aneurysmal dilatation) had the lowest TZo and TZe along the length of the artery and was significantly different compared to adjacent patches. This suggests that collagen is becoming activated and the tissue is transitioning into a collagen-dominated behavior at lower stresses. The earlier collagen activation may be due to the elastin being depleted in the area of the aneurysm compared to adjacent regions as the collagen replaces it to provide mechanical support. This transition appears to be related to the position/geometry of the aneurysm. As previously noted, position 3 generally was the most dilated area of the AAA. The outliers for TZo and TZe in position 3 and 4 originated from the same patient which presented with position 4 associated with the largest diameter.

Histological samples allowed for the analysis of aortic wall structure and composition as it relates to the mechanical properties of the aorta. More than 50% of these specimens presented non-discernible media, indicative of high degeneration and loss of structural integrity in the aortic tissue under investigation. Elastin, smooth muscle cells, and proteoglycans contents were assessed in the remaining samples. While elastin has been reported to be reduced in aneurysms [17], with ruptured AAAs presenting lower elastin content than non-ruptured ones [41], the regional heterogeneity of elastin content has not been previously reported. Present results showed no effect of the position along the length of the aorta on the content of medial constituents, but a significant effect of the circumferential position was found. Elastin and proteoglycans were observed to be present in inverse

proportions with smooth muscle (i.e., an increase in relative elastin and proteoglycans is associated with a decrease in smooth muscle). Of note, specimens collected from the anterior regions of the aorta presented significantly lower elastin content compared to posterior regions. As previously discussed, the only other parameter significantly affected by the position along the circumference was the LTM that characterizes the elastin-dominated mechanical behavior of the tissue at low strain. Greater elastin content and higher LTM in posterior regions of the aorta indicate an asymmetric remodeling and degenerative process at the wall, with asymmetric growth and thrombus accumulation occurring predominantly in the anterior bulging.

Literature findings suggest that the AAA pathophysiology is a multifactorial process, with cellular mechanisms integrally involved in the structural and functional changes that occur in the aortic wall. However, relatively few studies have investigated the heterogeneity of inflammatory response in human AAA and its possible implications.

An effect of circumferential position was previously reported by Hurks et al., who found that lateral regions of AAA exhibited increased inflammatory activity in the adventitia compared to anterior and posterior regions [16]. In contrast, the present study showed no significant variation in inflammatory cell count among circumferential regions, although it should be noted that different inflammation markers were analyzed. However, when the effect of axial position was investigated, tissue samples showed significant heterogeneity in terms of inflammatory infiltrates. Higher inflammatory cell activity (CD4+, CD8+) in the adventitia was found in samples collected from regions in position 4 compared to position 2 and 3. The localized increase in helper T-cells (CD4+) and cytotoxic T-cells (CD8+) together in this region suggests the presence of CD4+ T-cell phenotype Th1, which has been implicated in ECM degradation [42]. CD4+ T-cells were also reported to release cytokines that stimulate angiogenesis and fibroblastic collagen accumulation in the adventitia. Therefore, their appearance is highly indicative of aortic wall remodeling [43,44]. Interestingly, position 4 exhibited distinct mechanical behaviour compared to position 2 and 3, with a significantly higher TZo and TZe, possibly a result of increased fibrotic collagen deposition in the adventitia in combination with a degraded media.

These results are significant in the overall discussion on AAA risk assessment: the axial heterogeneity in inflammatory cells seems to have direct functional implications that can be seen in the biomechanical results of this study.

While the ex vivo mechanical and histological characterization of the aorta is essential for a thorough understanding of the structural and functional changes linked to AAA progression, the in vivo assessment becomes central when clinical application and disease management are the goal.

From in vivo, image-based regional assessment of the AAA population, the TAWSS was found to be affected by the position along the axis of the aortic geometry with areas of altered, recirculating flow (low TAWSS) marking the central aneurysm regions as a consequence of dilatation. The shear stress resulting from the viscous nature of blood flow is unlikely to load the aortic tissue to the point of failure; however, the wall-shear stress is involved in the processes of mechano-sensing and mechano-transduction responsible for the local pathological remodeling and structural degeneration of the aortic tissue. Because of the tissue's response to site-specific hemodynamic conditions, the aortic wall presents highly heterogeneous material properties, especially in the presence of degenerative processes. The effect of wall-shear stress has been linked to disease progression before, with literature reporting on its role in thrombus formation and accumulation—likely leading to hypoxia and further loss in tissue integrity, as well as aneurysm rupture [2,4–7]. The present findings, in agreement with previous literature, highlight the aneurysm as a region of significantly disturbed flow where further degeneration of the tissue is likely to occur. As the aorta dilates, its hemodynamics is subject to additional disturbance driven by geometrical changes, making the dilatation itself a region more and more prone to local pathological processes. Interestingly, the TAWSS was found to be significantly lower for the regions characterized by tissue specimens showing no discernible media (more than

50% of the histological samples), further highlighting the relationship between local fluid dynamic patterns and tissue degeneration.

On the one hand, the study of the local hemodynamics provides an understanding of the biological substrate for the aortic wall that leads to pathological remodelling and enlargement. On the other hand, the local deformability of the wall can help further characterize the state of regional weakening of the aortic wall and its propensity for rapid dilatation or rupture. The in vivo three-dimensional strain analysis allowed for the non-invasive assessment of localized wall behavior directly from dynamic CT imageswithout assumptions made on constitutive models. The region-averaged distribution of maximum principal strain for the AAA population further supports the concept of heterogeneity in aneurysmal tissue as a result of the heterogeneous remodeling, and the significant effect of the position along the length of the aorta on its biomechanics. The central regions corresponding to the aneurysmal sac (position 3, 4, 5) presented significantly smaller strain compared to the more proximal (neck) regions.

As the in vivo local deformability of the aorta relates to local mechanical properties, a large strain may be a possible consequence of flow impingement on the aortic wall identified in regions with a corresponding high TAWSS. Similarly, in the regions of low TAWSS, such as the larger diameter areas, the deformation of the aortic wall is expected to be small, especially in presence of a thick thrombus buffering the aorta. Therefore, in these regions, a larger strain may be indicative of intrinsic localized weakening of the tissue, as the low blood velocities are unlikely to be causing the deformation of the tissue. Of note, the distribution of strain presented more variability than the distribution of wall shear stress further highlighting the dual influence of loading (flow impingement) and material weakening on the localized wall strain.

This is a pioneering work on the regional heterogeneity of the AAA that comprises both in vivo and ex vivo analyses. The comprehensive investigation highlighted novel results on the heterogeneity of the aortic tissue along both the circumference and length of the aneurysm. The heterogeneity along the circumference of the aorta was only reflected in the ex vivo biaxial response at low strains, which was linked to variations of elastin content. A particularly marked effect of position along the aortic length, in contrast, was consistently observed in several ex vivo and in vivo properties, with the central regions corresponding to the aneurysmal dilatation (particularly position 3) being strikingly distinct from the adjacent patches (i.e., showing significant difference with respect to the other identified regions). The aneurysm area was characterized by disturbed hemodynamics likely to drive the pathological remodeling that results in a changed mechanical response as observed in the heterogeneity of collagen-related behavior (HTM, TZo, and TZe) and inflammatory markers content along the length of the artery.

First, the consistency observed between ex vivo and in vivo properties clearly suggests that in vivo biomarkers can eventually improve aneurysm assessment and outcome prediction, with regional heterogeneity providing a direction for future work on non-invasive risk assessment. Second, while the central regions of the aorta corresponded to the position of the aneurysmal sac and, therefore, included the location of maximum diameter, the present research clearly points out the shortcomings of the use of maximum diameter. The aortic size alone cannot fully characterize the regional variability and heterogeneity of the AAA tissue, especially when considering that rupture has been reported to often occur away from the location of maximum diameter [4,7].

This study presents limitations that need to be addressed. The sample size was limited, in terms of both patients and AAA tissue specimens; present results would benefit from analyses performed on a larger cohort that can better represent the variability among different patients and aortic anatomies. A larger study cohort would possibly allow for the investigation of difference in female versus male AAA patients, as well as age-matched analysis, given the effect of sex and aging on the arterial structure and function. Similarly, access to healthy aortic tissue and imaging would provide a means for valuable considerations.

There are shortcomings in the colorimetric analysis performed on histologies as only three colors on a pentachrome stain were analyzed: darker pixels may be recognized as black, ultimately affecting the elastin content results. Moreover, due to the scarcity of specimens containing measurable collagen in the media, along with the faint nature of the Musto/Movat staining for collagen, analysis for this constituent was not feasible. Further studies should be conducted to include Picrosirus Red staining to allow for collagen visualization, and Total Collagen Assay and ELISA to allow for quantitative assessment of collagen in the tissue. Additionally, the use of multiphoton microscopy would prove useful to investigate collagen fiber morphology (i.e., fiber thickness, direction, ondulation) as it relates to the mechanical properties of the tissue. While the Musto/Movat staining provided an insight into the elastin content of the specimens, it did not allow for any inference on the state of the elastin fibers, thus additional analysis would prove useful to access information on elastin fragmentation. Similarly, a look into extracellular matrix breakdown and matrix metalloproteinase would provide information on the proteolytic process associated with the degradation of the aortic media.

The IHC analysis had limitations due to the presence of background staining that can lead to a misinterpreted and overestimated cell count. In this regard, manual cell counting can be prone to errors and may be insufficient as a stand-alone metric for inflammation as cytokines, signaling interactions and molecular mechanisms are also involved in the complex inflammatory process.

The assumption of rigid aortic wall for CFD simulations also presented limitations. Despite being non-realistic, this assumption is an acceptable simplification that allows the characterization of the main hemodynamic patterns given the unknown, and extremely heterogeneous, patient-specific material properties that would be needed for more computationally expensive fluid–structure interaction (FSI) simulations.

While the primary objective of the present study was the ex vivo and in vivo mechanical characterization of the aortic tissue and its heterogeneity, the sparse experimental data did not allow a one-to-one regional correspondence for each analysis, therefore limiting the investigation of relationships among the different parameters. Future work will look at prospective longitudinal studies to assess regional aortic growth as a result of localized weakening with the aim of correlating regional growth to non-invasive parameters that can be measured clinically.

## 5. Conclusions

The workflow and methodology described herein allowed for the characterization of the aortic tissue with a comprehensive location-specific analysis and provided a distinctive insight on the assessment of AAAs. The effect of position on ex vivo and in vivo properties was investigated and showed the significant heterogeneity present in AAA tissue along both the length and the circumference of the aorta, with the most striking and consistent results being found with respect to the position along the length of the artery.

Present findings highlight the heterogeneity of AAAs and the essential role of regional characterization in the context of aortic assessment for disease management purposes. More work should be done towards the implementation of novel approaches accounting for the heterogeneity of the aneurysm—through correlation of the heterogeneity at the tissue level with non-invasive measurements—in order to improve risk stratification and clinical outcomes for individual AAA patients.

**Author Contributions:** Conceptualization, E.S.D.M.; Data curation, A.F.; Formal analysis, A.F., M.N., T.S. and L.N.; Funding acquisition, E.S.D.M.; Investigation, A.F., A.I. and A.B.; Methodology, A.F., T.S. and E.S.D.M.; Project administration, E.S.D.M.; Resources, R.D.M. and E.S.D.M.; Supervision, E.S.D.M.; Validation, A.F.; Visualization, M.N. and T.S.; Writing—original draft, A.F., M.N., T.S. and L.N.; Writing—review and editing, A.F., M.N., A.I., T.S., L.N., A.B., R.D.M. and E.S.D.M. All authors have read and agreed to the published version of the manuscript.

**Funding:** This research was funded by the Werner Graupe International Fellowship in Engineering and the Libin Cardiovascular Institute of Alberta scholarship, with contributions from the Natural

Science and Engineering Research Council of Canada Discovery Grant RGPIN/04043-2014, and RGPIN/07178-2019, and the Heart and Stroke Foundation Grant in Aid G 170019141.

**Institutional Review Board Statement:** The study was conducted according to the guidelines of the Declaration of Helsinki, and approved by the University of Calgary Conjoint Health Research Ethics Board (CHREB) (Ethics ID: REB15-0777).

**Informed Consent Statement:** Informed written consent was obtained from all subjects involved in the study.

**Data Availability Statement:** The data presented in this study are available on request from the corresponding author.

**Acknowledgments:** The authors would like to acknowledge the work done by Richard Beddoes, Flavio Bellacosa Marotti, and Christi Findlay in support of this research study.

**Conflicts of Interest:** E.S.D.M., R.D.M., and A.F. are co-founders and shareholders of the start-up company ViTAA.

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
