# Peer review of "Heterogeneity of Ex Vivo and In Vivo Properties along the Length of the Abdominal Aortic Aneurysm"

_applsci, doi:10.3390/app11083485_

Round 1

Reviewer 1 Report

In this paper by Forneris et al, the author’s have investigated the heterogeneity of abdominal aortic aneurysms (AAA) along the length, based on longitudinal and cyclical parameters. To assess regional differences along the length of AAA they have used in vivo imaging and modelled the wall shear stresses and undertaken principal strain analysis. In addition, they have taken ex vivo tissue and subjected it to biaxial testing, and also immunohistochemical staining to interrogate inflammatory cells in the tissue.

Overall, these are useful studies and provide helpful insight into potential difficulties in the clinical management of AAA. However, I do have some suggestions and comments as highlighted below, with my main area of concern being the histological analysis of the tissue.

Materials and Methods

  1. Figure 1; Line 104-111 – Definition of RP, LP, RA and LA in figure legend would be helpful
  2. Figure 1; Line 104-111 – it would be helpful in the figure legend to explain what ‘11’ and ‘22’ relate to.
  3. Line 152 – sentence missing some words after 0.05N? e.g. ‘was applied’.
  4. Line 191 – Reference missing for LTM and HTM being related to elastin and collagen respectively.

Immunohistochemical Analysis; Line 206+:

  1. Were formalin-fixed paraffin embedded sections taken, and what thickness sections were cut?
  2. Were aneurysms cut in cross section?
  3. Antibody manufacturer and working concentrations should be included.
  4. What secondary antibodies were used?
  5. What controls were run?
  6. Line 213/214 – it is unclear what e.g. H15 X15 and H20 x 20 refer to – please elaborate in the text
  7. Were histological sections sub-divided into the four different quadrants (RP, LP, RA and LA) as with in vivo and ex vivo measurements?
  8. It is stated that positively stained cells were counted in hotspots. This is going to be biasing the data, and a better measure would be a more randomised approach to choosing regions where the cells were counted. Were the hotspots located along a similar region of the wall for each AAA? If so, this should be stated.
  9. I feel that this study is not complete, without histological collagen and elastin analysis. The authors state that LTM and HTM are caused by elastin-based and collagen-dominated responses, but looking at this in the tissue would add weight to the paper. As the authors have access to tissue from these aneurysms this should be included. Simple image-based analysis would be sufficient e.g. looking at elastin breaks in the media using a stain such as elastin van gieson, or using picrosirius red for collagen. If possible, analysis of collagen fibres anisotropy could be performed with e.g. multiphoton microscopy.
  10. Furthermore, to look at extracellular matrix breakdown, matrix metalloproteinases (e.g. MMP-2 and -9 that have been shown to be upregulated in aneurysms - see Maguire et al (2019) Pharmaceuticals; 12(3), 118) could be looked at to see how these change biaxially.

  1. Line 231 – How was the geometry of the aorta reconstructed? Was this following standard procedures?

Results

  1. Line 288, Line 484: The authors state that the neck region was considered a non-aneurysmal control. Is this tissue actually ‘normal’ being that it is very close to regions that become aneurystic. Whilst I appreciate that getting true healthy abdominal aorta tissue is very difficult, have the authors any evidence that the tissue is similar to healthy tissue and doesn’t undergo changes?
  2. What was the average length of each of the regions characterised? Was this uniform for each patient?
  3. It is stated in the methods that 92% of patients gave consent were male, which means only 1 patient was female. Were there any differences in any parameters looked at for this patients AAA, as it is known that males have a greater propensity to develop AAA. Perhaps this could be commented on in the results or discussion.
  4. Markers of Inflammation: Line 358 –. Could some representative images of the staining be included?
  5. A subgroup of 7 AAA underwent IHC analysis. 6 main regions were analysed, yet 46 aortic samples were collected. Were some AAA divided more than 6 times, or analysed twice?
  6. For histological analysis, were the AAA divided into the four different quadrants (RP, LP, RA and LA). If so, were there differences between areas?

Discussion

Line 468, Line 509, Line 536 – Multiphoton microscopy, or Sirius red staining for collagen, or IHC staining for collagen types, would provide further evidence that there are differences in collagen fibres in different regions.

Reviewer 2 Report

This paper describes the comprehensive analyses for the region-specific properties of the tissue from the abdominal aortic aneurysm (AAA). The analyzed parameters included the biaxial mechanical properties and immunohistochemical analyses for CD4+ T cells, CD8+ T cells and CD68+ macrophages for the excised AAA. The analyses also included the computational in vivo assessment for the wall shear stress and the wall strain. The results demonstrated the parameter-specific heterogeneity along the length or the circumference of the aorta.

Major concerns

This study employed a unique approach in analyzing the heterogeneity of the AAA walls both in ex vivo and in vivo contexts. Overall, the methods and the results are clearly described. However, the discussion needs much improvement to draw a better understanding of the wall properties of AAA with clinical implications.

This study was aiming at obtaining better understanding of the progression and the rupture of AAA by comparing the parameters from different analytical modalities. Currently, the discussion is constructed with the summary of the individual findings and the literature survey. The discussion on the progression of AAA is largely speculative rather than based on their own data. Because the strength of this study is the multiple parameters from the same set of samples using different analytical modalities, relationships among the parameters should be determined, and their pathological and clinical implications needs to be discussed. In addition, because the authors are discussing the mechanical properties of the AAA walls in relation to elastin and collagen fibers, histological analysis should be performed for these extracellular matrix components, and their relationships to the mechanical properties should be analyzed.

Minor concerns

  1. For HTM11 and HTM22, please specify which one is for the longitudinal value and another for the circumferential value.
  2. Several data are mentioned only in the text without presenting in Figures. Please include the data for longitudinal LTM, circumferential CD4/CD8-positive cell contents, and circumferential/longitudinal CD68 contents. Likewise, please show the circumferential analyses for TAWSS and MPS. These would be essential to make relationships among different parameters.
  3. For immunohistochemical analysis, please show the representative images for the immunostaining, and show how the adventitial and medial layers were discriminated.
  4. It would be great if the authors could analyze the non-aneurysmal aorta even with limited numbers. Currently it is unclear whether the local heterogeneity of the aortic walls is specific to AAA or intrinsic to the aortic wall regions irrespective of the presence of AAA.

Reviewer 3 Report

The study by Fornesis et al. investigated the structural and mechanical properties of human aneurysmatic aorta along the length and the circumflex in 6 aortic regions. Wall-shear stress and three-dimensional principal strain analysis were performed in vivo. Biaxial testing and immunohistology were used by ex vivo studies. The authors found a significant heterogeneity of AAA tissue along both the length and the circumference of the aorta. The study suggests that the aortic size could not fully characterise the AAA tissue and the risk of rupture. The correlation of the heterogeneity at the tissue level with non-invasive measurements should be performed individually for a better risk stratification.

This study is very important, informative and it is of clinical relevance.

Comments:

  1. The patient characteristics should provide more information about patient’s comorbidities (diabetes, hypertension etc.).
  2. Were all AAA of atherosclerotic origin or inflammatory aneurysms were also included in the study?
  3. It would be important to demonstrate representative immunohistological stains which show differences in the expression and localisation of inflammatory cells in aortic segments.
  4. Elastin expression is referred to an unpublished Master-Thesis. Could you provide the data about elastin in this manuscript?
  5. The citation of the studies performed by the other groups could be expanded.

Reviewer 4 Report

Thank you very much for your paper, it could have important clinical consequence.  

Author Response

Thank you for your review and feedback. The results and discussion have been expanded and improved in the revised manuscript (INTRODUCTION – lines 67-73; METHODS – lines 215-225; RESULTS – lines 388-416 & 489-493; DISCUSSION – lines 604-622).

Round 2

Reviewer 1 Report

Although there are still a few limitations re. histological assessment of tissue, the authors have successfully justified their reasons not to include the data, and highlighted these, where necessary, in the manuscript.

Reviewer 2 Report

The authors have adequately addressed the concerns of this reviewer.